# Characterization of *CK2*, *MYC* and *ERG* Expression in Biological Subgroups of Children with Acute Lymphoblastic Leukemia

**DOI:** 10.3390/ijms26031076

**Published:** 2025-01-26

**Authors:** Luca Lo Nigro, Marta Arrabito, Nellina Andriano, Valeria Iachelli, Manuela La Rosa, Paola Bonaccorso

**Affiliations:** 1Cytogenetic-Cytofluorimetric-Molecular Biology Lab, Center of Pediatric Hematology Oncology, Azienda Ospedaliero Universitaria Policlinico-San Marco, 95123 Catania, Italy; marta-arrabito@hotmail.it (M.A.); nellinaandriano@yahoo.it (N.A.); valeria.iachelli@gmail.com (V.I.); larosa_manuela@libero.it (M.L.R.); paolabonaccorso@libero.it (P.B.); 2Center of Pediatric Hematology Oncology, Azienda Ospedaliero Universitaria Policlinico-San Marco, 95123 Catania, Italy

**Keywords:** acute lymphoblastic leukemia, children, *CK2*, *MYC*, *ERG*

## Abstract

Despite the excellent survival rate, relapse occurs in 20% of children with ALL. Deep analyses of cell signaling pathways allow us to identify new markers and/or targets promising more effective and less toxic therapy. We analyzed 61 diagnostic samples collected from 35 patients with B- and 26 with T-ALL, respectively. The expression of *CK2*, *MYC* and *ERG* genes using Sybr-Green assay and the comparative 2-ΔΔCt method using 20 healthy donors (HDs) was evaluated. We observed a statistically significant difference in *CK2* expression in non-HR (*p* = 0.010) and in HR (*p* = 0.0003) T-ALL cases compared to HDs. T-ALL patients with *PTEN*-Exon7 mutation, *IKZF1* and *CDKN2A* deletions showed high *CK2* expression. *MYC* expression was higher in pediatric T-ALL patients than HDs (*p* = 0.019). Surprisingly, we found *MYC* expression to be higher in non-HR than in HR T-ALL patients. *TLX3* (*HOX11L2*)-rearranged T-ALLs (27%) in association with *CRLF2* overexpression (23%) showed very high *MYC* expression. In B-ALLs, we detected *CK2* expression higher than HDs and *MYC* overexpression in HR compared to non-HR patients, particularly in *MLL*-rearranged B-ALLs. We observed a strong difference in ERG expression between pediatric T- and B-ALL cases. In conclusion, we confirmed *CK2* as a prognostic marker and a therapeutic target.

## 1. Introduction

Substantial advances have been made in the past five decades in the treatment of patients with acute lymphoblastic leukemia (ALL), the most common malignancy in childhood. In contemporary ALL treatment regimens, patients are stratified into different risk groups based on clinical and biologic characteristics at presentation as well as on early treatment response [1]. Improved understanding of the biologic heterogeneity of ALL and the development of sensitive minimal residual disease (MRD) response-monitoring techniques have improved modern risk stratification for childhood ALL [1]. The majority of patients with B-lineage ALL have somatic aneuploidy or recurrent chromosomal translocations, many of which have prognostic significance [2]. Conversely, the clinical significance of most recurrent genomic alterations in T-lineage ALL is less clear, and current clinical trials stratify patients with T-ALL based on their response to 8 days of prednisone administration or detection of MRD during the induction phase [3,4]. Despite excellent survival, exceeding 90%, relapse occurs in 15% to 20% of children with ALL, which remains the first cause of death in children with cancer [2]. Early relapses of both B- and T-ALL are a current challenge because of the high rate of second-line treatment failure. For this reason, analyses of significantly enriched signaling pathways might help to identify new markers for a better patient stratification and/or targets for a potential future tailored therapy. Pathways including *WNT/β-Catenin*, p53 and *PI3K/AKT/PTEN* with *ERG* overexpression may contribute to the dysregulation of kinase signals such as *Casein Kinase* 2 (*CK2*), which results in resistance to kinase inhibitors [5,6,7,8]. *CK2* overexpression has been observed in hematological malignancies such as acute and chronic leukemias, including T- and B-ALL and acute myeloid leukemia (AML) [6,8]. Elevated levels of *CK2α* are highly correlated with poor clinical outcome [8]. An important *CK2* target is *IKAROS* (*IKZF1*), a transcription factor that displays crucial functions in the hematopoietic system and controls the development in early B- and T-cells. Mutations that lead to reduced *IKZF1* function or expression have also been found to be a major genetic feature in human B-ALL [9] and loss of *IKZF1* also promotes the development of T-cell lymphoma/leukemia in mice, which suggests that *IKZF1* acts as a tumor suppressor in T- and B-ALL [9,10]. Indeed, despite its low prevalence, *IKZF1* loss of function is clearly a recurrent anomaly in human T-ALL [10]. Thus, *IKZF1* inactivation could play a causal role in disease progression in a subset of T-ALL cases, influencing the outcome [10]. Another proto-oncogene closely involved in many cancers, including leukemia, is *MYC*. Aberrant expression of *MYC* in leukemia results in an uncontrolled rate of proliferation and, thereby, a blockade of the differentiation process [11]. *MYC* is not activated by mutations in the coding sequence, but its overexpression in leukemia is mainly caused by gene amplification and aberrant regulation of its transcription [11,12]. A tightly controlled increase in *MYC* expression is required for differentiation, but prolonged and excessive *MYC* activity is oncogenic and leads to increased proliferation, altered adhesion and chromatin remodeling [12,13]. Synergistic interactions between *CK2* and *MYC* might lead to lymphocytes’ transformation; indeed, CK2 inhibitors, acting as an *IKZF1* activator, suppress *c-MYC* in an *Ikaros*-dependent manner in ALL cells [14]. It has been demonstrated that in adult ALL patients, high *c-MYC* expression correlates with clinical high-risk factors and high proliferation markers [14]. Another useful marker in pediatric leukemia is the Ets-related gene *ERG* (erythroblastosis virus E26-transforming sequence family member) which has an important role in early hematopoiesis and hematopoietic stem cell (HSC) maintenance [15]. *ERG* is preferentially and strongly expressed in the immature B- and T-lymphoid lineages, in addition to myeloid lineage cells. Previous studies suggest that *ERG* overexpression is associated with inferior clinical outcome [15]. In T-ALL patients, a high level of *ERG* expression has been associated with poor relapse-free survival [15]. Conversely, in childhood B-ALL patients, a higher white blood cell (WBC) count, higher relapse rate and poor relapse-free survival rate were associated with low *ERG* expression [16]. The expression of *CK2* in association with *MYC* and *ERG* has not yet been characterized in biological subgroups of pediatric ALL. The characterization of these genes would integrate new markers of disease and/or therapeutic targets in ALL. The cooperating activities of the selected genes are also supported by a recent article demonstrating that *ERG* and *c-MYC* coordinate a regulatory network in *BCR::ABL*1 B-ALL, controlling the expression of genes involved in several biological processes [17].

## 2. Results

### 2.1. Expression of CK2, MYC and ERG in T-ALL Cases

We preliminarily screened 26 patients with T-ALL for diagnostic molecular characterization and we found 5 cases with *PTEN*-*Exon7* mutation (19%), 3 with *PICALM/MLLT10* rearrangement (11.5%), 7 with *TLX*3 (*HOX11L2*) alteration (27%), 1 with mutated *TP53* (4%), 16 with *CDKN2A* deletion (61.5%) and 6 with *CRLF2* overexpression (23%); additionally, 3 out of 20 patients exhibited *IKZF1* deletion (15%), respectively (Appendix A). Firstly, we compared *CK2* expression in T-ALL cases to HDs and the CEM cell line: we observed a similarly high expression between T-ALL and CEM (Appendix A) and a statistically significant difference in T-ALL patients (more in HR than in non-HR cases) vs. HDs (Figure 1). We point out that 10 out of 11 HR-T-ALL patients (90%) presented *CK2* mRNA expression up to five times higher than that of HDs and two times higher than other T-ALL patients. Moreover, four out of five PTEN *Exon7* mutated cases showed a high *CK2* mRNA level (Appendix A). Furthermore, we correlated *CK2* expression among cases with *CDKN2A*, *IKZF1* deletions and *CRLF2* overexpression, respectively. We found out that children with *IKZF1* and *CDKN2A* deletions showed high *CK2* mRNA levels, although they were not statistically significant (Appendix A). We did not find any difference in *CK2* expression among cases with *CRLF2* overexpression compared with normal *CRLF2* expression (Appendix A).

*MYC* expression in the CEM-cell line could be used to perform an exact comparison with T-ALLs. We observed a statistically significant difference in MYC expression in children with T-ALL compared to HDs (Figure 2A): 13 out of 26 T-ALL patients were identified as *MYC-high* (50%), and of these, 5 were HR and 8 were non-HR, respectively. Seven out of eight *MYC-high* non-HR patients showed a *TLX3* (*HOX11L2*) rearrangement. Overall, we observed statistically higher *MYC* expression in non-HR T-ALLs than in HR T-ALLs (Figure 2B). In PTEN-*Exon7*-mutated patients, we observed low expression of *MYC* (Appendix A). Moreover, we compared *MYC* expression with *CDKN2A* and *IKZF1* status and we did not observe any statistically significant difference in MYC expression (Appendix A). By contrast, in T-ALL cases with high *CRLF2* expression, we detected very high *MYC* expression compared to cases with normal *CRLF2* expression (Appendix A). Five out of the six cases with *CRLF2* overexpression had a *TLX3* rearrangement.

*ERG* expression in the CEM cell line was very low, compared to HDs and T-ALL cases (Appendix A), but statistically significant different expression was observed in children with T-ALL vs. HDs (Figure 3A); this was more prevalent in HR than in non-HR cases. However, the difference in ERG expression between the two T-ALL subgroups was not statistically significant (Figure 3B). When we analyzed the five T-ALLs with *PTEN Exon7* mutations, they showed slightly higher levels of ERG mRNA than *PTEN Exon7* wild type patients (Appendix A).

Moreover, we studied thymocytes as normal internal control for *CK2*, *MYC* and *ERG* expression compared with HDs. CK2 expression was absolutely comparable between thymocytes and HDs. Instead, MFC MYC expression in thymocytes was comparable to T-ALL samples. This is due to an overlapping metabolic rewiring in activated T-cells and MYC-transformed lymphocytes, similar to that which occurs in tumor cells (with the activation of proto-oncogenes such as c-Myc) [16]. Erg expression was comparable in thymocytes and HDs, too (Appendix A).

### 2.2. Expression of CK2, MYC and ERG in B-ALL Cases

We screened 35 B-ALL patients, who were stratified into four biological subgroups according to the most common molecular alterations detected at diagnosis of pediatric B-ALL (Appendix A). We performed *CK2* expression analysis, observing a statistically significant difference in CK2 expression in B-ALL patients with respect to HDs (Figure 4A). The *CK2* expression was homogenous among the four selected biological ALL subgroups, as shown in Figure 4B.

As for CK2 expression, we also found a statistically significant difference in MYC expression with respect to HDs (Figure 5A). Interestingly, we observed a correlation between *MYC* expression with a prognostic genetic subgroup: in the Ph+ cases, *MYC* expression was high, whereas in *ETV6::RUNX1*, the rate of positive B-ALL was very low. In the latter subgroup, two cases, who subsequently presented a late relapse, showed a higher value of *MYC* expression. Moreover, in seven out of seven HR patients in the “B-others” subgroup, we detected high *MYC* expression; this was also observed in four cases included in the *MLL*-R B-ALL subgroup (Figure 5B).

In cases with B-ALL, we also observed a high level of *ERG* expression compared with that of HDs (Figure 6A). These values were higher than *ERG* expression findings in T-ALLs. Among different biological subgroups, *ERG* expression was higher in *ETV6::RUNX1*-positive ALL patients than in *BCR::ABL1*, B-others and, in particular, *MLL*-R B-ALL patients (Figure 6B).

This inverse correlation with clinical risk was confirmed by the fact that 84% of HR B-ALL cases showed low expression of *ERG*, as we report in Table 1, which depicts the *CK2*, *MYC* and *ERG* expression according to the *Final risk group* in children with T- and B-ALL. Instead, *CK2*, *MYC* and *ERG* expression according to genetic alterations is summarized in Table 2.

## 3. Discussion

Due to the high-risk disease characteristics and significant toxicity associated with chemotherapy, the outcome for ALL patients is less encouraging for defined subgroups of patients, including both standard and high-risk groups [18]. Novel targeted therapies offer the promise of effective anti-leukemic activity with reduced toxicity, but given the different molecular and genetic alterations that occur in ALL patients, it is improbable that a single agent will be effective for all patients. For this reason, it is mandatory to identify patient-specific therapy [18]. Moreover, some subtypes of this condition, such as T-ALL, despite the numerous genetic aberrations involved, lack reliable markers for precise stratification and/or potential targeted therapy [19]. Based on these considerations, we focused on studying three important genes involved in the main metabolic pathways of pediatric ALL [7]. The expression of *CK2* in association with *MYC* and *ERG* overexpression has not been characterized in biological subgroups of pediatric ALL, yet. Here, we demonstrated that *CK2* is overexpressed in pediatric ALL, resulting in a potential marker and, moreover, in a stable therapeutic target. Both T- and B-ALLs show higher levels of *CK2* expression compared with HDs. Moreover, in T-ALLs with HR features, we showed a median *CK2* expression up to five times higher than that of HDs. *CK2* is a regulatory serine/threonine kinase which is ubiquitously expressed, and its activity is required for activation of pro-survival pathways [20]. Several pathways (WNT/β-Catenin, p53 and PI3K/AKT/PTEN with ERG overexpression) may contribute to the dysregulation of kinase signals such as *CK2*, which results in resistance to kinase inhibitors [8]. The importance of phosphorylation makes protein kinases and phosphatases promising therapeutic targets for a wide variety of human disorders [6]. Several studies demonstrated that different panels of leukemia patients and cell lines are sensitive to *CK2* inhibition by TBB (tetrabromobenzotriazole) and CX-4945, a small molecule that is orally bioavailable, as shown by Song C et al. This hints that CX-4945 exerts its anti-leukemic effect via inhibition of *CK2*-mediated phosphorylation [21]. These data support the use of CX-4945 in a phase I clinical trial for treatment of hematologic malignancies [21,22] but since *CK2* overexpression is a hallmark of ALL, the cytotoxic potential of CX-4945 in T-cell and B-cell ALL was also elucidated [13]. Overexpression of *CK2* appears to impart a survival advantage in cancer cells by suppressing apoptosis through its action on a variety of cellular and nuclear substrates and favoring cell growth. Within *CK2* targets, an important role is played by the transcription factor *IKZF1* that acts as a tumor suppressor in T- and B-ALL; genome-wide analyses have shown that 30% of pediatric B-ALL and approximately 5% of T-ALL present a deletion or dysfunction of the *IKZF1* gene [9,10]. *IKZF1* haploinsufficiency is sufficient to mediate leukemogenesis. *CK2* phosphorylates *IKZF1* and impairs its function as a tumor suppressor in leukemia models [21]. Indeed, molecular and pharmacological inhibition of *CK2* restored *IKZF1* function as a tumor suppressor [13,22].

In our study, *IKZF1*, a prognostic factor in pediatric B-ALL [23], was also found to be mutated in children with T-ALL (3 out of 20 cases). This finding was recently confirmed by a retrospective analysis among adult and pediatric patients with T-ALL [10]. Moreover, it was recently demonstrated that *CK2* and *IKZF1* regulate chemoresistance to doxorubicin via repression of *BCL2L1* (BCL-XL) and a combination treatment involving a CK2 inhibitor and doxorubicin has a synergistic therapeutic effect on high-risk B-ALL in vivo [24]. It is also known that *MYC* is an *IKZF1* target gene [8]; thus, CK2 inhibitor acts as an *IKZF1* activator and suppresses *MYC* expression in an *IKZF1*-dependent manner in ALL cells [8]. For this reason, we focused on *MYC* expression in our pediatric patients, and we demonstrate here, for the first time, a correlation between *MYC* overexpression, *TLX3* (*HOX11L2*) rearrangement and *CRLF2* overexpression in T-ALL patients. *MYC-high* expression was seen in seven *TLX3* (*HOX11L2*) rearrangements (seven out of eight non-HR T-ALL patients); this was previously correlated with a poor prognosis [25,26]. Moreover, five out of seven non-HR T-ALL patients showed signs of *CRLF2* overexpression, which has also been demonstrated as a poor prognostic marker in children with T-ALL [27]. These findings confirmed that *TLX3* expression is not a prognostic indicator in pediatric T-ALL and that high levels of *MYC* expression are broadly present in T-ALL [28]. In patients with *PTEN*-*Exon7* mutation, as expected, we observed a low expression of *MYC.* Conversely, we showed that in pediatric B-ALL patients, there is an evident correlation between a *MYC-high* profile and poor prognostic genetic subgroups. Interestingly, patients with 11q23-R or *MLL/KMT2A*-rearranged B-ALL show an extremely high level of *MYC* expression. Accordingly, it has recently been shown that the proliferation of *MLL/KMT2A*-rearranged B-ALL cells was decreased upon *MYC* depletion and that MYC-protein abundance in *MLL/KMT2A*-rearranged B-ALL cells was much higher than in non-*MLL/KMT2A*-rearranged B-ALL cells [29]. To date, this B-ALL subtype is correlated with a very poor prognosis, especially in infancy, and it is associated with a high rate of relapse due to a high grade of chemoresistance [30], soliciting the use of new targeted drugs. The oncogenic potential of the transcription factor *ERG* induced us to determine the characterization of *ERG* expression in pediatric ALL. Our data hint at *ERG* overexpression in pediatric ALL patients compared with HDs. In particular, we identified high *ERG* expression as an independent adverse prognostic factor in children with high-risk T-ALL and found that it was associated with an expected inferior outcome [5]. The *ERG* gene is involved in signal transduction pathways that regulate and promote cell differentiation, proliferation and tissue invasion [6,7]. Previous studies suggest that *ERG* overexpression is associated with inferior clinical outcome [15,16]. In T-ALL patients, a high level of *ERG* expression has been associated with poor relapse-free survival (RFS) [15]. It is also demonstrated that co-activation of the *PI3K/AKT* pathway and *ERG* overexpression collaborate with a lack of *PTEN* and prostate-specific androgen response in the development of prostate carcinoma [31]. Functional assays revealed that *ERG* may modulate kinase signaling pathways, but there are no data showing a direct correlation between *CK2* and *ERG* expression. *ERG* overexpression resulted in dephosphorylation of AKT(Ser473), suggesting that *ERG* overexpression represses *AKT* activation [6,7]. So, it has been proposed that *ERG*-driven drug resistance overrides PI3K/AKT signaling by alternative pathways which need to be further investigated for effective drug design and adapted therapies for *ERG*-overexpressing high-risk leukemias [6,7]. Conversely to T-ALL, we found low *ERG* expression in HR patients with B-ALL, hinting that it plays the opposite role to the transcription factor within the cell. This was confirmed in a very recent article showing that *ERG* can act as a tumor-suppressor gene or as an oncogene with the opposite results [32].

Based on our data, we outlined a patient-specific profile based on the expression of these three biomarkers (*CK2*, *MYC*, *ERG*) and the mutational screening for *PTEN-Exon7*, *IKZF1* and *TLX3* genes. The *CK2* overexpression associated with *ERG* overexpression (independently from *MYC* expression) related to other genetic alterations (*PTEN Exon7* mutation, *CDKN2A* or *IKZF1* deletion) could be helpful in the identification of a patient-specific profile. To be precise, we propose a “*CK2-plus* group” for T-ALL: *CK2-high*, *ERG-high*, *PTEN Exon7* mutation, *CDKN2A* and *IKZF1* deletion identify a subgroup of pediatric T-ALL patients with a very poor outcome, based on the occurrence of a high rate of relapse and/or death due to progressive disease. Despite the several molecular events that drive the different human leukemia subtypes, high *CK2* levels appeared as a common denominator in all of them, suggesting that targeting-*CK2* could represent a multi-potential therapeutic strategy [8]. Although our study presents several limitations regarding the number of children with ALL and retrospective analyses, our findings strongly suggest that *CK2* and *ERG* represent relevant molecular markers that generate a risk-adapted treatment strategy for high-risk patients with ALL, which must be confirmed by a prospective study in a larger population. Strikingly, a recent study showed the synergistic effect of CK2 inhibitor (CX-4945) and mTOR inhibitor (rapamycin), based on the evidence that the inhibition of CK2 enhances IKAROS activity as a repressor of mTOR [33]. Thus, this dual inhibition will enhance the anti-leukemic effect in patients with HR ALL, who showed, as we demonstrated, high expression of CK2, impaired activity of IKZF1 and disruption of the PI3K/AKT/mTOR pathway [33]. Moreover, in an even more recent report, it has been confirmed that CK2 inhibitor (CX-4945) strongly suppresses the expression of WD Repeat-Containing Protein 5 (WDR5) in T-ALL by restoring Ikaros gene function. This result confirms the interaction between CK2 and Ikaros, in association with WDR5, and supports a synergistic therapeutic intervention with targeted drugs [34]. In conclusion, based on our report as a preliminary project, we propose that a prospective study should be performed (as ancillary in the upcoming international AIEOP-BFM protocol on T-ALL) with the aim of identifying the “*CK2-plus* group” and subsequent treatment with CK2 inhibitors in case of MRD persistence; relevant treatment should take the form of an experimental targeted therapy.

## 4. Materials and Methods

### 4.1. Patient Samples

Our study included 61 patients: 26 with T-ALL and 35 with B-ALL. All of the included patients were diagnosed in our center from September 2000 to September 2016. The clinical follow-up ranged from 72 to 216 months.

The B-ALL cohort included the following biological subgroups: 8 patients with t(9;22)/chromosome Philadelphia-positive (Ph+)/*BCR::ABL1*-positive; 16 with t(12;21)/*ETV6::RUNX1*-positive; 4 with rearranged 11q23 (-R) *MLL/KMT2A*-positive; and 7 without known translocations defined as “B-others”, respectively. Based on protocol response stratification criteria, patients were classified as Standard Risk (SR), Intermediate Risk (MR), or non-HR (*n* = 17) and High Risk (*n* = 18) in B-ALL, or as HR (*n* = 11) and non-HR (*n* = 15) for T-ALL patients, respectively. Patients received treatment according to the ongoing protocols [AIEOP-BFM ALL2000/R2006 and ALL2009]. The expression levels of *CK2*, *MYC* and *ERG* genes were retrospectively analyzed in diagnostic cDNA samples. Biological and clinical features, including adverse events and outcomes, are shown in Table 3. Institutional review board approval and an informed consent from parents were obtained, according to protocol guidelines and our hospital’s regulations.

### 4.2. Cell Lines and Human Normal Bone Marrows

CCRF-CEM (T-ALL bearing *PTEN* and *CDKN2A* deletions) and JURKAT (T-ALL bearing a *PTEN* missense mutation and *CDKN2A* and *CREBBP* deletions) cell lines were used as positive controls for T-ALL. They were cultured in RPMI-1640 medium with 10% heat-inactivated fetal bovine serum, 1% L-glutamine and 1% Pen-Strep. REH (B-ALL bearing an *ETV6::RUNX1* fusion transcript) and MHH-CALL-4 (B-ALL bearing a *JAK2*-I682F mutation, constitutive phosphorylation of *JAK2* and *STAT5* and *CRLF2* overexpression) were used as B-ALL positive controls (ATCC). REH was cultured as mentioned above. Instead, MHH-CALL-4 was cultured in RPMI-1640 medium with 20% heat-inactivated fetal bovine serum, 1% L-glutamine and 1% Pen-Strep. Cells were kept at 37º C in 5% CO2 and split every 3 days.

Twenty samples of bone marrow (BM) from children without a history of malignancy (healthy donors for bone marrow transplantation) were selected and served as healthy donors (HDs). We collected informed consent from their parents prior to bone marrow donation. In addition, thymocytes were also analyzed as normal internal controls for *CK2*, *MYC* and *ERG* expression with respect to healthy donors (Appendix A).

### 4.3. RNA Isolation and RT-qPCR

Mononuclear cells from BM samples were isolated by Ficoll gradient centrifugation and cryopreserved. Total RNA was extracted using TRizol Reagent (Invitrogen, CA, USA) following the manufacturer’s protocols. First-strand cDNA was synthesized from total RNA with reverse transcriptase and random primers using the Superscript Reverse Transcriptase (Invitrogen, CA, USA). Samples were selected based on the quality and quantity of RNA (OD_260_/OD_280_ ratio: 1.8–2.0). Quantitative Real-Time RQ-PCR was performed with a 96-well optic plate using the QuantStudio 7 Flex (Applied Biosystem, Life Technologies). Each sample was tested in duplicate. Expression levels were normalized by *GUS* (endogenous control) and calculated using the comparative 2^−ΔΔCt^ method. The Comparative Cycle Threshold (CCT) method was used to determine the relative expression levels of *CK2*, *MYC* and *ERG*, using the median of ΔCt from HDs in two replicates and expressed as 2^−ΔΔCt^ (ΔCt = *GUS*-gene of interest). Each reaction mixture consisted of 1 μL of cDNA (from 1 mg of RNA), 7.5 μL of SYBR Green Universal Master Mix (Thermo Fisher Scientific, Waltham, MA, USA), 1 μL of each primer (250 nmol/mL) and 4.5 μL of deionized water. The PCR cycle started with an initial 95 °C, 10 min melt step, followed by 40 cycles of 15 s at 95 °C and 60 s at 60 °C. Primer sequences of *CK2*, *MYC*, *ERG* and *GUS* are shown in Appendix A. In addition, T-ALL patients were screened for the following molecular alterations: *PTEN-Exon7* mutations; *PICALM::AF10* fusion transcript; *TP53* and *pS6* mutations; *TXL*3 rearrangements and *CRLF2* overexpression, respectively.

### 4.4. Data Analysis

Patients were classified as having *high* or *low CK2* or *MYC* or *ERG* expression. We used the median value of gene expression fold changes as the cut-off. Statistical analysis was performed using GraphPad Prism 7 Software. The data are presented as the Mean ± Standard Deviations (SDs). A two-tailed *p* < 0.05 was indicative of a statistically significant difference between groups (Appendix A).

## Figures and Tables

**Figure 1 ijms-26-01076-f001:**
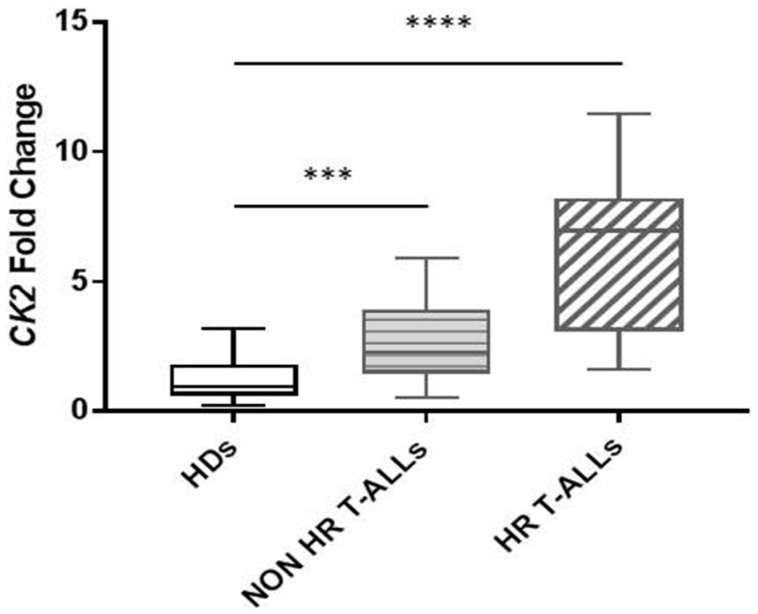
*CK2* expression difference in children with T-ALL: non-HR cases vs. HDs (mean CK2 fold changes 3.094 vs. 1.24, *p* = 0.0105 ***) and HR cases vs. HDs (6.223 vs. 1.24, *p* = 0.0003 ****).

**Figure 2 ijms-26-01076-f002:**
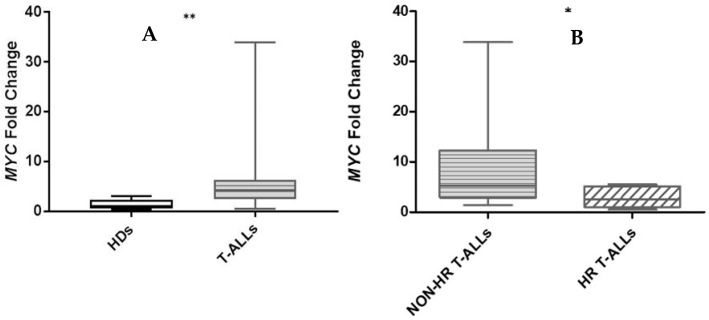
*MYC* expression in children with T-ALL vs. HDs (*p* = 0.019 **) was statistically higher in T-ALL than in HDs: mean fold change (MFC) 6.802 (range 0.475–33.855) vs. 1.280 (ranging from 0.298 to 2.966), respectively (**A**); *MYC* MFC expression was statistically different between non-HR and HR T-ALL patients (9.596 vs. 2.992, respectively *) (**B**).

**Figure 3 ijms-26-01076-f003:**
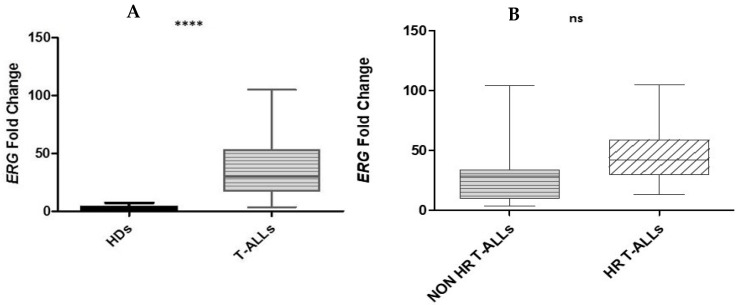
The *ERG* mRNA level in children with T-ALL vs. HDs was statistically significant (mean T-ALL 38.497 vs. mean HDs 2.605 ****) (**A**). The different *ERG* expression between the two T-ALL subgroups (non-HR mean 33.114 vs. HR mean 45.838) was not statistically significant (ns) (**B**).

**Figure 4 ijms-26-01076-f004:**
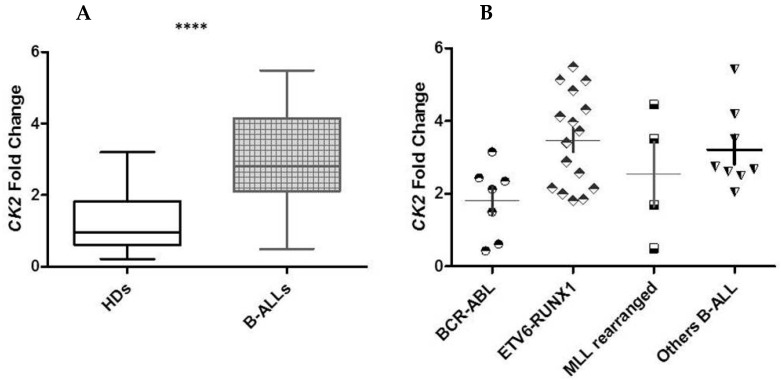
The *CK2* mRNA level was higher in B-ALL than HDs (a mean *CK2* fold change of 3.011 vs. 1.237, respectively) (*p* = 0.0003 ****) (**A**). The expression was homogeneous among the four cytogenetic subgroups (**B**).

**Figure 5 ijms-26-01076-f005:**
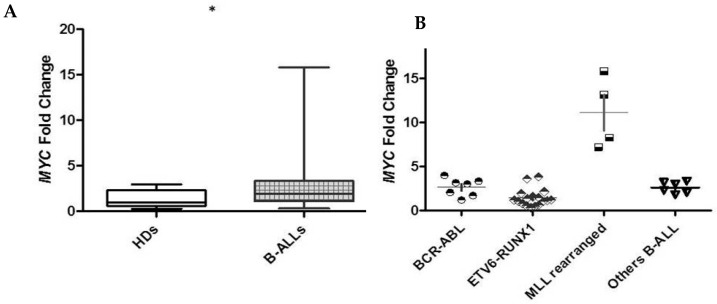
There was a statistically significant difference in MYC expression in B-ALL patients than in HDs (a mean fold change of 2.887 vs. 1.284, *p* = 0.005 *) (**A**); *MYC* expression among cytogenetic subgroups of B-ALL patients (**B**).

**Figure 6 ijms-26-01076-f006:**
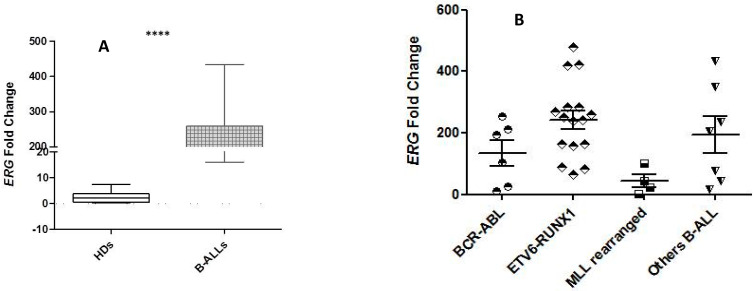
The *ERG* mRNA expression in B-ALL patients was significantly higher (****) compared to the expression value in HDs (a mean of 191.770 vs. a mean of 2.605 in HDs) (**A**); the *ERG* expression in biological subtypes of B-ALL: *ETV6::RUNX1* was higher than in other subgroups. In particular, *MLL-R* B-ALL showed very low ERG expression (MFC 54.054) (**B**).

**Table 1 ijms-26-01076-t001:** *CK2*, *MYC* and *ERG* expression according to the *Final risk group* in children with T- and B-ALL.

		*CK2* Expression		*MYC* Expression		*ERG* Expression	
		*Low* *n* (%)	*High* *n* (%)	*Low* *n* (%)	*High* *n* (%)	*Low* *n* (%)	*High* *n* (%)
T-ALL	Non-HR (*n* = 15)	9 (60)	6 (40)	7 (47)	8 (53)	12 (80)	3 (20)
	HR (*n* = 11)	1 (9)	10 (91)	6 (54)	5 (46)	2 (19)	9 (81)
B-ALL	Non-HR (*n* = 16)	7 (44)	9 (56)	12 (75)	4 (25)	6 (38)	10 (62)
	HR (*n* = 19)	7 (37)	12 (63)	2 (21)	17 (79)	16 (84)	3 (16)

**Table 2 ijms-26-01076-t002:** *CK2*, *MYC* and *ERG* expression according to genetic alterations in children with T- and B-ALL.

Genetic Alteration		Analyzed Patients	*CK2* Expression		*MYC* Expression		*ERG* Expression	
			*Low* *n* (%)	*High* *n* (%)	*Low* *n* (%)	*High* *n* (%)	*Low* *n* (%)	*High* *n* (%)
*BCR/ABL1*		B-ALL (*n* = 8/35)	3 (37)	5 (63)	1 (12)	7 (88)	6 (75)	2 (25)
*ETV6/RUNX1*		B-ALL (*n* = 16/35)	8 (50)	8 (50)	14 (88)	2 (12)	6 (38)	10 (62)
*MLL rearranged*		B-ALL (*n* = 4/35)	2 (50)	2 (50)	-	4 (100) °	4 (100) ^a^	-
*PTEN Exon7* Δ *or inactivating mutations*		T-ALL (*n* = 5/26)	2 (40)	3 (60)	5 (100)	-	1 (20)	4 (80)
*PICALM/MLLT10*		T-ALL (*n* = 3/26)	2 (67)	1 (33)	1 (33)	2 (67)	1 (33)	2 (67)
*TLX3/HOX11L2*		T-ALL (*n* = 7/26)	3 (43)	4 (57)	-	7 (100) °	5 (71.5)	2 (28.5)
*O* *T* *H* *E* *R* *S*	*CDKN2A Δ*	B-ALL (*n* = 2/7)	2 (100)	-	-	2 (100)	2 (100)	-
		T-ALL (*n* = 16/23)	8 (50)	8 (50)	7 (44)	9 (56)	8 (50)	8 (50)
	*IKZF1* Δ	B-ALL (*n* = 2/7)	2 (100)	-	-	2 (100)	2 (100)	-
		T-ALL (*n* = 3/20)	-	3 (100)	2 (67)	1 (33)	-	3 (100)
	*hi-CRLF2*	B-ALL (*n* = 2/7)	1 (50)	1 (50)	-	2 (100)	1 (50)	1 (50)
		T-ALL (*n* = 6/25)	3 (50)	3 (50)	1( 17)	5 (83)	4 (67)	2 (33)

^a^ Very low ERG expression; ° very high MYC expression.

**Table 3 ijms-26-01076-t003:** Clinical and biological features of 61 children with ALL analyzed for expression of *CK2-MYC-ERG*.

	T-ALL *n* = 26 *n* (%)	B-ALL *n* = 35 *n* (%)	*Total* *n* = 61 *n* (%)
*Gender*			
Female	6 (23)	22 (63)	28 (46)
Male	20 (77)	13 (37)	33 (54)
*Age at diagnosis (years)*			
1–9	10 (38)	27 (77)	37 (61)
≥10	16 (62)	8 (23)	24 (39)
*Presenting WBC count*/μL			
<10,000	4 (16)	12 (34)	16 (26)
10,000–50,000	3 (11)	11 (32)	14 (23)
50,000–100,000	8 (31)	2 (6)	10 (16)
≥100,000	11 (42)	10 (28)	21 (35)
*Prednisone response* ^			
Good	17 (65)	27 (77)	44 (72)
Poor	9 (35)	8 (23)	17 (28)
*MRD risk group* *			
Standard risk	3 (12)	9 (26)	12 (20)
Intermediate risk	14 (54)	19 (54)	33 (54)
High risk	4 (15)	7 (20)	11 (18)
Not performed	5 (19)	-	5 (8)
*Final risk group* §			
SR+MR (non-HR)	15 (58)	16 (46)	30 (49)
HR	11 (42)	19 (54)	31 (51)
*Event*			
No	18 (69)	24 (69)	42 (69)
Yes	8 (31)	11 (31)	19 (31)
*Outcome*			
Alive	19 (73)	28 (80)	47 (77)
DOC	5 (19)	4 (11)	9 (15)
DOD	2 (8)	3 (9)	5 (8)

Table legend. ^ Good: <1000 leukemic blasts/µL after 7 days of prednisone administration; poor ≥ 1000/µL. * Minimal residual disease (MRD). Standard risk: negative both at day + 33 (end of induction or phase Ia) and day + 78 (end of consolidation therapy of phase Ib). High risk: MRD level ≥ 10^−3^ at day + 78; Intermediate risk: all others (AIEOP-BFM ALL 2000–2009). § Patients with a *BCR::ABL1* or *MLL/KMT2A* rearrangement at diagnosis or a prednisone poor response (PPR) or ≥10% blasts (detected by cytofluorimetry) in bone marrow (BM) performed at day + 15 (for those who were enrolled at AIEOP-BFM-ALL 2009 protocol) or induction failure (≥5% blasts at day + 33 BM), were stratified into the HR final group independently from MRD results. The following events were considered: relapse or death of disease (DOD) or death of complication (DOC) during treatment.

## Data Availability

The original contributions presented in this study are included in the article/Appendix A. Further inquiries can be directed to the corresponding author.

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
