# Peer review of "Characterization of CK2, MYC and ERG Expression in Biological Subgroups of Children with Acute Lymphoblastic Leukemia"

_ijms, 2025, doi:10.3390/ijms26031076_

Round 1
Reviewer 1 Report
Comments and Suggestions for Authors
The authors try to find potential factors helping to better stratify childhood ALL for improvement the results of treatment. The selection of genes whose expression was studied in the ALL group seems to be promising. The article is very interesting, the study is well designed. The methods are clearly described.
However there are some issues that should be corrected or presised:
- there is no information about exact time when children were treated - it should be corrected. Probably it is a historic group of children given protocols of treatment.
- no clear treatment outcome and time of observation is marked
- it would be more clear if there was a table or a scheme how authors propose to identify subgroups (like in verse 351)
- there is no clear conclusion in the text- it should be corrected
- the introduction is not really adequate to nowadays prognosis in ALL, and the literature is not very recent. As I suppose it is because the data are not very recent
- in the line 91 there is "e" instead of "and"
Author Response
Point-by-point reply to the Reviewer #1
Dear Reviewer, thank you for your positive introductive general comment on our article.
Please accept our apologize if we send you these replies so late.
Comment 1. There is no information about exact time when children were treated – it should be corrected. Probably it is an historic group of children given protocols of treatment.
Reply. It is a right consideration. We decided to perform our analyses in selected biological subgroups, based on genetic data and minimal residual disease (MRD) findings, in order to evaluate the expression of these genes in the non-High Risk and HR subgroups. Please see the Materials and Methods, section 2.1, lines 81-82.
Comment 2. No clear outcome and time of observation is marked.
Reply. Please see the Materials and Methods, section 2.1, lines 82, for the time of observation and Table 1 for the outcome. We would highlight that we cited the incidence of event and the Death of complications, which would avoid with the use of targeted therapy, as we propose with our experiments.
Comment 3. It would be clearer (“more clear”) if there was a table or a scheme how authors propose to identify subgroups (like in verse 351)
Reply. The biological subgroups are defined by WHO classification. The other ones are defined by response to therapy based on MRD findings, inducing clinicians to treat patients with different intensive chemotherapy, as indicated by non-HR and HR subgroups.
Comment 4. There is no clear conclusion in the text – it should be corrected.
Reply. Please see Discussion. We finally propose to identify a “CK2-plus group”, mainly in children with HR-MRD T-ALL, who will probably benefit from an experimental targeted therapy.
Comment 5. The introduction is not really adequate to nowadays prognosis in ALL, and the literature is not very recent. As I suppose it is because the data are not very recent.
Reply. This is a right consideration. Actually, we took the time for experiments and for the clinical follow-up. Based on your comment, we modified the Introduction and update the literature.
Comment 6. In the line 91 there is “e” instead of “and”.
Reply. We apologize but we did not find this error. In lines 91-92 there are only “and”. Anyway, we checked the entire manuscript with the specific purpose to correct this type of error.

Reviewer 2 Report
Comments and Suggestions for Authors
In the manuscript Characterization of CK2, MYC and ERG Expression in Biological Subgroups of Children with Acute Lymphoblastic Leukemia, Luca Lo Nigro et al. Luca Lo Nigro et al. have provided insight into the molecular mechanisms of acute lymphoblastic leukemia (ALL) in children, especially the expression of CK2, MYC, and ERG. This approach is relevant for a better understanding the disease and potential personalized therapy.
The authors studied the expression of CK2, MYC, and ERG genes in different biological subgroups of ALL in children. They found a clear correlation between these genes, clinical characteristics, and genetic changes.
The paper highlights the potential of CK2 and ERG as therapeutic targets, which may contribute to the development of new treatments for high-risk patients.
Tables and graphs provide clear visual information, supporting key conclusions.
What should be worked on
· The study included only 61 patients, which may limit the generalizability of the results - A larger cohort would be desirable to confirm the findings to ensure statistical power and better generalizability.
· The work relies on retrospective analysis and does not include new experimental research, which limits innovation - the inclusion of new experimental findings, such as functional studies on cell lines, would enrich the work and increase its scientific value.
· The discussion of the role of ERG and CK2 in signaling pathways is not deep enough, especially regarding their mutual interactions and possible synergies in therapies - Deeper consideration of how CK2, MYC, and ERG interact in signaling pathways and how their inhibition could synergize in therapies. Adding signaling pathway diagrams and schematics could help better understand the complex mechanisms.
· A more detailed discussion of potential controversies or discrepancies in the literature related to the results is lacking
· It is necessary to thoroughly review the text to correct grammatical and stylistic errors.
Comments on the Quality of English LanguageIt is necessary to thoroughly review the text to correct grammatical and stylistic errors.
Author Response
Point-by-point reply to the Reviewer #2
Dear Reviewer, thank you for your positive introductive general comment on our article.
Please accept our apologize if we send you these replies so late.
Comment 1. The study included only 61 patients, which may limit the generalizability of the results. A larger cohort would be desirable to confirm the findings to ensure statistical power and better generalizability.
Reply. We completely agree with this comment. The final purpose of our report is to produce the basis for a prospective project as an ancillary study in the future international AIEOP BFM ALL protocol. Please see the Discussion section on rows 330-333.
Comment 2. The work relies on retrospective analysis and does not include new experimental research, which limit innovation – the inclusion of new experimental findings, such as functional studies on cell lines, would enrich the work and increase its scientific value.
Reply. We started our project from deeply analyzing the expression of the selected genes in cell-lines: CEM and Jurkat for T-ALL; REH and MHH-CALL4. Considering the expression of these genes, we also paid the right attention to the normal counterpart of T- and B-lineage ALL. We also performed Phospho-flow cytometric analyses of STAT3-STAT5-CREB-PTEN-pS6, producing preliminary findings which we decided to not include in this manuscript. As we mentioned above and in the manuscript, we are planning to propose our project as ancillary study in the upcoming international protocol, considering the present article as a starting point.
Comment 3. The discussion of the role of ERG and CK2 in signaling pathway is not deep enough, especially regarding their mutual interactions and possible synergies therapies – Deeper consideration of how CK2, MYC and ERG interact in signaling pathways and how their inhibition could synergize in therapies. Adding signaling pathway diagrams and schematics could help better understand the complex mechanisms.
Reply. CK2 aberrations and its inhibition is widely discussed in acute and chronic leukemias. MYC expression is related to TP53 and other modifications of the cell cycle in several tumors, including acute leukemias. ERG is still under investigation because of its impact on outcome (see IKZF-plus positive children with B-lineage ALL). Since the selected genes were analyzed separately as single ones, we decided to not drawing any scheme in order to avoid a misleading figure.
Comment 4. A more detailed discussion of potential controversies or discrepancies in the literature related to the results is lacking.
Reply. As mentioned above, CK2 is the most treated gene in respect of MYC and ERG. We propose a method of patients’ identification in order to treat them with a targeted therapy (CK2 inhibitors) and to support a proposal of new classification for T-ALL, based on MYC and ERG. In the literature, the correlation of specific genes in children with ALL is lacking.
Comment 5. It is necessary to thoroughly review the text to correct grammatical and stylistic errors.
Reply. The manuscript has been sent to an internal author service for improving the English language.
